# Biocontrol of Aflatoxins Using Non-Aflatoxigenic *Aspergillus flavus*: A Literature Review

**DOI:** 10.3390/jof7050381

**Published:** 2021-05-12

**Authors:** Rahim Khan, Farinazleen Mohamad Ghazali, Nor Ainy Mahyudin, Nik Iskandar Putra Samsudin

**Affiliations:** 1Department of Food Science, Faculty of Food Science and Technology, Universiti Putra Malaysia, Serdang 43400, Malaysia; sirifrahim1@yahoo.com (R.K.); nikiskandar@upm.edu.my (N.I.P.S.); 2Department of Food Service and Management, Faculty of Food Science and Technology, Universiti Putra Malaysia, Serdang 43400, Malaysia; norainy@upm.edu.my; 3Laboratory of Halal Science Research, Halal Products Research Institute, Universiti Putra Malaysia, Serdang 43400, Malaysia; 4Laboratory of Food Safety and Food Integrity, Institute of Tropical Agriculture and Food Security, Universiti Putra Malaysia, Serdang 43400, Malaysia

**Keywords:** aflatoxins, biocontrol, non-aflatoxigenic *Aspergillus flavus*, biotic and abiotic factors

## Abstract

Aflatoxins (AFs) are mycotoxins, predominantly produced by *Aspergillus flavus*, *A. parasiticus*, *A. nomius*, and *A. pseudotamarii*. AFs are carcinogenic compounds causing liver cancer in humans and animals. Physical and biological factors significantly affect AF production during the pre-and post-harvest time. Several methodologies have been developed to control AF contamination, yet; they are usually expensive and unfriendly to the environment. Consequently, interest in using biocontrol agents has increased, as they are convenient, advanced, and friendly to the environment. Using non-aflatoxigenic strains of *A. flavus* (AF^−^) as biocontrol agents is the most promising method to control AFs’ contamination in cereal crops. AF^−^ strains cannot produce AFs due to the absence of polyketide synthase genes or genetic mutation. AF^−^ strains competitively exclude the AF^+^ strains in the field, giving an extra advantage to the stored grains. Several microbiological, molecular, and field-based approaches have been used to select a suitable biocontrol agent. The effectiveness of biocontrol agents in controlling AF contamination could reach up to 99.3%. Optimal inoculum rate and a perfect time of application are critical factors influencing the efficacy of biocontrol agents.

## 1. Introduction

Aflatoxins (AFs) are secondary metabolites produced by *Aspergillus flavus*, *A. parasiticus*, *A. nomius*, and *A. pseudotamarii* [1,2]. AFs are organic compounds with lower molecular weight, typically produced by fungal mycelia and accumulated in conidia and sclerotia. AFs contaminate a wide range of crops, including corn, oilseeds, rice, and nuts [3,4,5,6]. AFs contamination in cereals may occur during pre- or post-harvest stages [7,8]. Hot temperature and high humidity stimulate fungal growth in fields and storage. Contamination by AFs is responsible for substantial commercial losses throughout the world [9,10,11]. AFs are among the most toxic compounds that adversely affect humans and animals’ health [12,13,14,15,16,17]. AFs are mutagenic, teratogenic, genotoxic, and carcinogenic compounds, causing severe diseases in humans, poultry, fishes, and cattle under long-term exposure [18,19]. AFs can penetrate the feed and food chain, posing a threat to even newborns [20,21]. While several AFs were currently identified, AFB_1_, AFB_2_, AFG_1_, and AFG_2_ are the four most significant AFs. The IARC (International Agency for Research on Cancer) classifies AFB_1_ as the most toxic, mutagenic, and Group 1 human carcinogen [22,23,24], causing chronic and acute diseases in children and the elderly. AFB_1_ carcinogenicity has long been linked to the liver; however, recent epidemiological studies revealed that it was also carcinogenic to the pancreas, kidney, bone, bladder, and central nervous system [25,26,27,28]. According to El-Serag [29], Bruix et al. [30], and Yoshida et al. [31], AFB_1_ exposure could increase the hepatocellular carcinoma (HCC) risk for up to 30 times, particularly in those who infected with hepatitis B virus. The inhalation of dust contaminated by AFB_1_ may cause tumors in humans’ respiratory tracts [32]. Furthermore, AFB_1_ disturbs the cytochrome P450 enzymes involved in steroid production [33]. Sometimes, AFB_1_ could get into the blood–testis barrier, resulting in spermatogenesis disorder [34]. Based on toxicological syndromes, AFs contamination can be divided into acute aflatoxicosis and chronic aflatoxicosis.

Acute aflatoxicosis is distinguished by a high-dose exposure of AFB_1_ for a short time, causing hepatotoxicity [35,36]. Acute aflatoxicosis is characterized by vomiting, fever, liver injury, pulmonary or cerebral edema, anemia, necrosis, diarrhea, kidney failure, and fatigue [37,38]. Several incidences of acute aflatoxicosis are reported in India, Malaysia, and Kenya [39,40,41,42]. In contrast, chronic aflatoxicosis is a low-dose exposure for a long duration, causing cancer and other severe diseases in humans. Some research reported that chronic exposure to AFB_1_ caused the deaths of 250,000 people in Africa and China [43,44,45]. Human exposure to AFs can be direct or indirect. The inhalation of AFB_1_-contaminated dust is an excellent example of direct exposure to AFs, resulting in the tumor in the human respiratory tract. On the other hand, the intake of AF-contaminated milk (AFM_1_) and other dairy products carried over contaminated feed is indirect exposure to AFs. The consumption of eggs and animal meat contaminated by AFs is another example of indirect exposure to AFs. Kaplan et al. [46] estimated human average intake of AFs at around 10–200 ng/kg per day. Humans’ health risks related to contaminated food consumption are becoming a serious problem all over the world. Countries where strict rules for AFs are not implemented, resulting in high health risks related to AFs exposure. Therefore, every country should implement strict rules for AF levels in their food products [47].

## 2. Global Distribution of *Aspergillus flavus* and Aflatoxins

*Aspergillus* section *Flavi* contains the most prevalent aflatoxigenic fungi, including *A. flavus* and *A. parasiticus*. The less prevalent aflatoxigenic species in this section are *A.nomius*, *A. pseudotamarii*, *A. bombysis*, and *A. parvisclerotigenus* [48]. *Aspergillus* species are remarkably different in AF production; some are aflatoxigenic while others are non-aflatoxigenic [49,50]. Alternatively, *A. flavus* is the most common species in crops producing AFs, and cyclopiazonic acid and non-aflatoxigenic strains are rare [50,51,52]. *A. flavus* can be found in decaying vegetation, crops, and seeds as a saprophyte or parasite. Soil is the main source of primary inoculum responsible for infection in crops vulnerable to AF contamination. The infection of *A. flavus* on the aerial parts of crops is different, depending on their rhizosphere habits [53]. The hot and humid weather and the absence of suitable storage facilities are favorable conditions for the growth of *A. flavus* and AF production [54]. For instance, tropical and subtropical regions with climate change encourage AF^−^ producing *A. flavus* to produce AFs in large quantities [55]. Corn and peanuts are the only crops consumed by humans worldwide, and unfortunately, highly vulnerable to AF contamination [56]. Around 40% of the loss of productivity due to infections caused by AFs has been increased in many developing countries [6].

## 3. Factors Affecting Aflatoxin Production

Several physical (abiotic) and biological (biotic) factors influence fungal growth and AF production [57]. In crops, AF contamination occurs during harvest, as the weather is wet due to unseasonal rains. Moreover, insect damage, drought, and heavy rainfall favor fungal growth. The degree of mycological infiltration and AF contamination varies with time and region [58]. Nature depends on fungal strains [59,60] and other microbes’ interference, moisture content, temperature, and resulting soil conditions. Fungal spores can enter through either damaged pod walls, insects, or pollination.

Additionally, nutrient deficiency in plants may increase AF levels. Recent studies showed high levels of AF production at 25 °C to 28 °C [61,62]. Likewise, high humidity (83–88%) and optimal CO_2_ and O_2_ have been found to influence fungal growth and AFs production [63]. Alternatively, lower concentrations of CO_2_ and O_2_ may inhibit fungal growth and AFs production. The presence or absence of certain compounds and elements can also control the AFs production, such as glucose, sucrose, and fructose that provide a suitable environment for fungal growth, while cadmium and iron slower down the fungal growth and AF production [64,65].

Similarly, climate change could significantly influence the AF^+^ life cycle, changing host–pathogen relationships and host resistance. It could directly impact the ability of AF^+^ species to produce AFs and their overall resilience [66,67]. Climate change not only affects host–pathogen relationships in specific areas but also promotes the emergence of new diseases and modifications in fungal biodiversity caused by fluctuations in their ecological niches [68,69,70]. Certain AF^+^ species are declining in one environment and reappearing in other regions because of climate change. The ability of AF^+^ species to adapt to such environmental changes can be perceived by continuously evolving combinations of AFs in food and feed.

## 4. Aflatoxin Management

Researchers are actively involved in preventing AFs production and spread, as the dangers of AFs to livestock and human health cannot be underestimated. Several pre- and post-harvest prevention measures, such as good agricultural practices, including deep plowing, manuring, irrigation, and maintaining water supply to the crops, are considered the best options for reducing AF contamination in crops [71,72,73]. Recent studies have suggested that irrigation in the late season could increase soil moisture contents and reduce the soil temperature, resulting in a decreased AF levels in crops [74]. These physical strategies, however, are not always feasible [19]. Apart from physical methods, various chemical strategies have been used for several years to lower AF levels in foods and feeds [75,76]. While almost all emphasis has been focused on controlling AFs contamination in crops, the most effective method is to use ammonia [77,78].

Additionally, fungicides such as amphotericin B, voriconazole, posaconazole, caspofungin, and voriconazole are effective against *A. flavus* invasion and AF contamination during pre-harvest stages. However, there is the risk of potential environmental pollution and health issues from fungicides [79,80]. Therefore, it is necessary to eliminate the risk by replacing chemical fungicides used with eco-friendly methods.

Crop varieties resistant to AFs are produced by breeding and genetic engineering techniques, yet no suitable resistant variety has been commercially developed [81]. Similarly, AF decontamination in food is convenient, but it is expensive and challenging [82]. Therefore, an interest in using biological control strategies has been developed, as they are helpful, friendly to the environment, and natural opponents of AF^−^ producing strains of *A. flavus* [83,84,85]. These strategies exploit some microorganisms’ antagonistic effects, such as bacteria, yeasts [86], and AF^−^ strains [87], on the development and production of AFs produced by AF^+^ strains. It has been reported that lactic acid bacteria such as *Bacillus* subtilis effectively inhibit the growth of various molds [88]. The inhibition is usually caused by competition for space and available nutrients needed for AFs biosynthesis but not for AF^+^ strains by co-existing microorganisms.

Similarly, *Flavobacterium aurantiacum* has been found to remove AFs from different foodstuffs. Likewise, *Pseudomonas* helps develop a healthy root system by its rapid colonization of the rhizosphere, stimulating plant defense mechanisms resulting in plant resistance to pathogens [89,90,91,92,93]. Faraj et al. [94] demonstrated that both *B. subtilis* have inhibitory effects on *A. flavus* and AF production growth. Mixing *B. subtilis* with groundnut diminished the deleterious effects of *A. flavus* on groundnuts. Mishagi et al. [95] have reported a 60–100% reduction in *A. flavus* incidence in synthetic media when treated with *P. cepacia* bacteria. Kong et al. [96] examined the potential antifungal activity of *B. megaterium* against the growth of *A. flavus* in groundnut kernels in vitro and in vivo. However, it has been found that biological control of AFs using AF^−^ strains is more productive compared to bacterial strains [97,98]. Therefore, biological control strategies based on AF^−^ strains could be viable options for reducing pre-harvest AF contamination in crops. The efficacy of AF^−^ strains are based on their stability and aggressiveness against AF^+^ strains [99,100]. Thus, this study focuses on the recent developments in the use of AF^−^ strains in reducing AF contamination in crops.

## 5. Advantages of Biocontrol of Aflatoxins Using Non-Aflatoxigenic *Aspergillus flavus*

Biocontrol methods are more effective and innovative to control AF contamination in crops. The application of biocontrol agents (AF^−^) carries some adaptations in fungal populations, which persist throughout the food chain. These adaptations prevent the grains from AF contamination during storage and transport; even environmental conditions are favorable for fungal growth. In biocontrol methods, the application of AF^−^ strain in the field remarkably reduces AF contamination in crops [101,102]. Similarly, like air, AF can disperse *Aspergillus* spores-communities, improve safety within the treated, and positively affect neighboring fields [103]. The positive impacts of AF^−^ strains can benefit crops and other plants for several years. This means a single dose of AF^−^ strain could benefit the treated crop and the second season crop, which missed the treatment [104].

## 6. Selection of Non-Aflatoxigenic Strains

Biocontrol is a promising method to reduce AF contamination in crops. Recent studies reported reducing AF contamination by applying AF^−^ strain to the soil around growing plants. When the crop is vulnerable to fungal attack during drought conditions, these AF^−^ strains competitively exclude the AF^+^ strains in the soil and reduce AF concentrations. Dorner [105] reported the reduction in AF contamination in a cornfield using AF^−^ strains. In other research, Dorner [105] assessed the efficacy of AF^−^ for AFs control in peanuts. AF^−^ strains can be found in air, soil, and plants. Usually, both AF^+^ and AF^−^ strains mutually occur in different ecosystems. The ability of AF^−^ strains competing with AF^+^ strains for nutrients provides an opportunity to use them as biocontrol agents. Different techniques have been developed to discover the suitable AF^−^ strain for biocontrol use. Some of them are based on phylogenetic features, while others on phenotypic characteristics such as sclerotial size. Based on sclerotial morphology and production, *A. flavus* can be divided into two distinct morphotypes, including S-strain and L-strain. The S-strains produce a large number of small-sized sclerotia (>400 µm in diameter), whereas the L-strains produce a small number of large-sized sclerotia (<400 µm in diameter). Moreover, S-strains produce a higher concentration of AF compared to L-strains. Molecular techniques may describe the phylogenetic relationships between *A. flavus* strains successfully. Several polymerase chain reaction (PCR)-based pyrosequencing methods are currently being developed to detect genes responsible for AF production and discover suitable biocontrol agents [106]. Abbas et al. [107] isolated some AF^−^ strains, including K49, F3W4, NRRL 58,974, NRRL 58,976, and NRRL 58,988. The classification was based on their growth rate, pigmentation, fluorescence, and AF production.

## 7. Efficacy of Non-Aflatoxigenic Strains as Biocontrol Agents

AF^−^ strains have been suggested as biocontrol agents in the hope that they would inhibit the growth of AF^+^ and thereby reduce AFs contamination. Previous studies conducted by Erhlich [108] revealed that co-inoculation of AF^−^ strains with AF^+^ substantially reduced the production of AF in corn under in vitro conditions. The potential for biocontrol of AFs using AF^−^ strains has been demonstrated under field conditions in cotton [109], peanuts [85], and corn [97,110]. These scientists have applied the AF^−^ strain to the soil as infested grain cultures of barley, rice, or wheat, whereas [111] inoculated corn ears directly by injection. In the cotton studies performed by Cotty [98], the AF^−^ strains were failed to suppress AFs contamination when it was sprayed on the cottonseed immediately before the bolls formed but were effective when sprayed on the soil later.

Similarly, a study conducted by Abbas [63] has demonstrated that soil inoculation of AF^−^ strain (K49) with AF^+^ strain (F3W4) mixture significantly reduced AFs contamination (74–95%) in corn. The degree of AF reduction found in his analysis was similar to the reductions obtained in other studies, using soil inoculation of corn and other crops. In Georgia, different studies have reported reductions in AF levels (80–87%) in cornfields after using AF^−^ strains against AF^−^ producing strains of *A. flavus*. Likewise, in cotton fields, the application of AF^−^ strain has decreased the amount of AFB_1_ from 75% to 99.8% [112]. Furthermore, Dorner et al. [99] reported the reduction of AFs concentrations between 74.3% and 99.8% in peanut crop when they applied the AF^−^ strains with non-aflatoxigenic strains of *A. parasiticus*. Peanuts produce fruiting bodies below the soil and hence increase the chances of biological control of AFs.

In another study, Dorner et al. [99] reported a 10–100 times increase in propagule density of the *Aspergillus* community when they co-inoculated the mixture of non-aflatoxigenic strains of *A. flavus* and *A. parasiticus* with AF^+^ strains. Additionally, their research has shown that *A. flavus* strains were more dominant over *A. parasiticus* in the displacement of AF^+^ strains in the soil. Dorner et al. [87] noted that the application of AF^−^ strains to the soil would control soil-borne infection and AFs contamination in crops like peanuts; however, the same treatment in some crops like corn will be difficult. On the contrary, an AF^−^ strain (CT3) was tested for its efficacy in AFs reduction, but it does not show effectivity like K49 to mitigate AFs contamination in corn. On the other hand, Cotty and Mellon [113] noted that co-inoculation of AF36 (AF^−^ strain) with AF^+^ ultimately displaced AF^+^ strain and markedly reduced AFs contamination in cottonseed.

Moreover, Chang et al. [114] identified an AF^−^ strain (TX9-8) by screening subgroups of AF^−^ strains. Co-inoculation of TX9-8 strain with AF^+^ strain with 1:1 ratio reduced AF production. No reduction in AF concentration has been observed when TX9-8 was injected one day later in AF^+^ strain. This competitive exclusion was possibly due to the vigorous growth of TX9-8 against AF^+^ strain [115]. Recently, Atehnkeng et al. [116] found La 3279 as the most efficient strain, decreasing AF contamination by >99.3%. Similarly, Ehrlich et al. [117] found the same results regarding secalonic acid reduction when they co-inoculated AF^−^ stain with *Penicillium oxalicum*. They assumed that the two co-inoculated species might cause competition for energy (ATP) required for the biosynthesis of secondary metabolites. There is an assumption that AF^−^ strains competitively exclude the AF^+^ strains when co-inoculated, resulting in the reduction of AFs contamination in crops [118]. Although AF^−^ strains have been employed to minimize AF infections in crops, the mechanism of AF^−^ strains’ intervention on AF^+^ strains remains unknown [119,120,121,122].

## 8. Factors Affecting the Efficacy of Biocontrol Agents

### 8.1. Inoculation Method

For many years, AF^−^ strains have been used on cornfield soil. Although the use of K49 in the soil can reduce AF levels by 65% [123], the direct use of AF^−^ strain on corn ears is immensely more efficient. A clay-based water-dispersible granule system was also developed to spray AF^−^ strain on corn silk directly. This management decreased AF production by up to 97%.

### 8.2. Inoculum Rate

Inoculum concentration is an essential factor for the effective control of AF contamination. Recent studies have revealed a direct relationship between the inoculum rate and AF’s efficacy^−^ strain in decreasing AF concentrations [124]. Studies demonstrated a significant reduction in AF concentration in peanuts when AF^−^ inoculum increased from 2–50 g/L. In the USA, research was conducted in which an AF^−^ strain (NRRL 21,368) with different quantities (0, 2, 10, and 50 g) was applied to the cornfield [125]. The AF levels for whole kernels were 337.6, 73.7, 34.8 and 33.3 μg/kg for the above quantities. Other research showed AF concentrations of 718.3, 184.4, 35.9 and 0.4 μg/kg in corn kernels, which demonstrated 74.3%, 95.0% and 99.9% of AF reduction. In the following years, the retreated field with AF^−^ strain showed a significant reduction in AF levels. According to Pitt and Hocking [119], the same results were achieved when tested in Australia.

### 8.3. Optimal Time for Non-Aflatoxigenic Strains Application

Research showed that with the concentration of AF^−^ strains, the time of its application significantly affects their efficacy. The application of AF^−^ strain at earlier stages significantly reduced AF levels in cotton. Similarly, Kabak and Dobson [126] suggested the co-inoculation of AF^+^ and AF^−^ strains (TX9-8) to reduce the AF contamination; however, if the AF^−^ strain is applied one day later, AF^+^ strains, fewer or no reduction in AF concentrations will be achieved.

### 8.4. Abiotic Factors

The time for the application of AF^−^, depends on the significant environmental conditions. Abiotic factors such as water activity and temperature directly affect AF^−^ strains’ efficacy by controlling spore germination, hyphal growth, and spore-production [127].

#### 8.4.1. Water Activity and Growth of Non-Aflatoxigenic Strains

Water plays a vital role in all biological practices. The main factor is the ambient water availability instead of the overall water content inside the hosts with microbes. The water content accessible to microbes in substrates is known as water activity. In substrates, water activity and total water content are interrelated. This helps to quantify the actual water content and microbial growth on the substrate. The respiration rate of AF^−^ strain used for reducing AF needs water. The water content in food performs an essential role in the growth of fungi and other biological activities. Once water availability is low, food spontaneously attains biological safety since it reduces the decomposition process through respiration. Seasonal variation and high humidity result in water availability for food, providing a breeding place for fungi. Moist is the primary source of crop losses [128], as the water content in grains increases fungal invasion. *A. flavus* grows at high water content (175 g/kg) and low temperatures (10–15 °C). As soon as the water content plunges from 175 g/kg to 94 g/kg, *A. flavus* cannot persist at 30–40 °C, demonstrating the significance of water content to the growth of fungi. Recent literature has shown that most of the molds could not propagate at a relative humidity of less than 70% [129]. Maintaining a lower water activity in preserved seeds, particularly in tropical regions, could be more challenging; hence, seeds containing high moisture content should be dried before storage to uphold seed sustainability against the fungal activity.

#### 8.4.2. Temperature and Growth of Non-Aflatoxigenic Strains

Microbial growth is exceptionally conspicuous in tropical and subtropical regions, where high temperatures and humidity prevail in most areas. High temperature and humidity favor fungal growth. Species like *A. candidus* have higher thermal tolerance, growing even in hot temperatures. However, some *Aspergillus* species show vigorous growth at a lower temperature (10–20 °C) [130]. Since *Aspergillus* does not reproduce at a higher temperature, the grains could be well-preserved at 40 °C [131]. Reed et al. [132] reported temperature as the primary factor in deciding field sustainability for mold. Under laboratory conditions, *A. flavus* multiplies when the temperature is around 10 °C [133]. However, no *A. flavus* growth occurred in the field environment if the temperature was less than 20 °C. Alternatively, Pitt and Hocking [119] reported faster growth for A. parasiticus at 15 °C under laboratory conditions, while, in field environments, their growth started at 17–20 °C. Thus, the application of AF^−^ strain should be delayed until the field temperature reaches 20 °C [134].

### 8.5. Biotic Factors

Low temperature and high water content in storage provide favorable conditions for insects, mites, and other microorganisms to grow. Insects’ respiration process produces hot spots in seeds, causing grain charring that affects seed quality and germination. In grains, insects’ activities increase the surrounding bulk’s temperature and water content, providing favorable mold growth conditions. Studies have shown that seeds damaged by insects are highly susceptible to fungal contamination [135]. Some fungi absorb insects and boost their populace, while others repel pests by secreting harmful toxins. Magan [136] reported that other microorganisms and environmental conditions significantly influenced the growth of AF^+^ strains, AF production, and competitiveness. Insects and mites are carriers as they carry fungal spores in their bodies. Studies have shown that mite infections supplement the *A. flavus* growth, as they carry fungal spores to fresh grains. Magan [136] suggested that mites are secondary vectors, carrying fungal spores into infected grains. In infected grains, mites take the fungal spores and carry them into their bodies or digestive tracts. When mites enter the fresh grain, they inoculate the fungal spores in it. The study discovered that mites seek out preferred fungi and digest a more significant percentage of their spores. Thus, these mites’ heavy infestations can be linked to damage from the mites and fungi associated with them. Some mites are growth inhibitors for fungi too. Some *Aspergillus* species are abundantly found on *Acarus siro*, indicating the symbiotic relationship of the fungi with their preferred mites.

### 8.6. Physiological Manipulation of Non-Aflatoxigenic Strains

Most of the fungal niches are not persistent as they modify their features according to the external environment [137]. In unfavorable environments, xerophilic fungi produce small polyols, which allow their enzymatic systems to work efficiently. Similarly, *A. flavus* accumulates glycerol and erythritol in their conidia during unfavorable conditions [138]. Therefore, fungal propagules used for biocontrol must be resistant to environmental stresses [120]. According to Magan [136], agricultural management could improve the resistive performance of biocontrol agents.

Furthermore, sugar and polyol mixture could boost spore germinability in severe environmental conditions. The conidia, which have a high amount of glycerol and trehalose, grow quicker than other conidia. Likewise, Abadias et al. [139] indicated high resistivity of *Candida sake* to water stress as their spores contain a high concentration of glycerol and erythritol. Gasch [137] suggested a link between environmental changes and adaptation length and proposed a conidial adjustment time. Thus, a strong adaptation with a short modification time makes biocontrol agents more competitive under critical conditions.

## 9. Conclusions

The above review showed that AF contaminates many cereal crops throughout the world. AFs producing molds, including *A. flavus*, contaminates these commodities at different stages within the food web. The strategies and tools developed for AF analysis have their advantages and disadvantages. Despite the immense information controlling AF, contamination continues with its harmful effects on human health, agro-industry, trade, and financial growth. This issue becomes more severe as AF’s contaminated cereal crops are essential for most of the world population. The AF’s contaminated crops are used in foods and feed products, resulting in many severe diseases in humans and animals. Thus, in every country, consumers and animals are persistently at risk. *Aspergillus* studies on their environmental conditions and the central perspective of farming systems can develop new AF control equipment. The pre-harvest methods (fungal population ecology, reproduction, and gene manipulation) are suitable for AF control; still, attention must be given to the environmental effects affecting these practices. For instance, the AF^−^ strains sometimes worsen AF’s issues by getting AF^−^ producing genes during vegetative fusion or sexual reproduction. This can be prevented by DNA-DNA hybridization to fully understand the genetic structure of AF^−^ strains and detect gene deletions in their chromosomes.

Furthermore, ear rot-resistant corn breeding might be a safe option for AF control, but it could take many breeding seasons due to AF^−^ resistant genes’ polygenic characteristics. The study of gene role and expression in different environmental conditions is necessary to understand the host-induced ecological reactions. Some resistant varieties are not fully adapted to grow in the field and are susceptible to AF contamination. Biocontrol techniques are more effective, environment-friendly, and economical for reducing AF in crops. The use of biocontrol agents brings some changes to the fungal communities that remain throughout the food chain. These changes prevent AF contamination during storage and transport; even environmental conditions are favorable for fungal growth. The application of biocontrol agents in the field remarkably reduces AF levels in crops from harvest until use. As *Aspergillus* spores can be dispersed by air, these fungal communities improve safety within treated fields and positively impact the neighboring fields, which means that a single dose of AF^−^ strain could benefit the treated crop and the second season crop that missed the treatment.

## Data Availability

Not applicable.

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
