# Peer review of "Biocontrol of Aflatoxins Using Non-Aflatoxigenic Aspergillus flavus: A Literature Review"

_jof, 2021, doi:10.3390/jof7050381_

Round 1

Reviewer 1 Report

Aflatoxin contamination is a serious problem worldwide but specially in countries with tropical climate, limited food resources and where storage of foodstuffs is challenging. Reduction of aflatoxin production by environment friendly biocontroll agents are optional in countries where people are dependent on natural resource of water, and non refined foods uncontrolled by legislations.  To my understanding biocontrol is the only option and every new aspect is important. 

 Question 1: line 483, The authors claim that AF- strains may even worsen the AF situation by getting AF producing genes by sexual recombination.  How can this be prevented of monitored?

Question 2: line 114, What increases susceptibility to what, sorry for stupidity?

Author Response

Comments and Suggestions for Authors

Aflatoxin contamination is a serious problem worldwide, but especially in countries with a tropical climate, limited food resources, and foodstuffs' storage is challenging. Reduction of aflatoxin production by environment-friendly biocontrol agents is optional in countries where people are dependent on the natural resource of water and non-refined foods uncontrolled by legislations.  To my understanding, biocontrol is the only option, and every new aspect is important.

Question 1: line 565, the authors claim that AF- strains may even worsen the AF situation by getting AF-producing genes by sexual recombination.  How can this be prevented or monitored?

Ans: This can be prevented by DNA-DNA hybridization to fully understand the genetic structure of AF- strains and detect the gene deletions in their chromosomes

Question 2: line 158, what increases susceptibility to what, sorry for stupidity?

Ans: Line 158,  the sentence has been modified and rewritten as follow;

“The above information indicated that AFB1 exposure resulting in impairments in cellular immunity (reducing phagocytic activity or T cell number) and hence reducing the host resistance to infections.”

Note: The first section of the manuscript has been rewritten as suggested by the third reviewer due to which the line number is changed.

Submission Date

16 March 2021

Date of this review

02 Apr 2021 10:13:13

Reviewer 2 Report

GENERAL COMMENTS:

The paper deals with an interesting topic of clear interest for the readers of Journal of Fungi. The paper presents a review of the biocontrol of Aflatoxins using non-aflatoxigenic A. flavus.  The review presents an adequate structure and an adequate literature study (recent and valid). The use of English language is correct and the text is easy to read and understand.

Overall, I suggest to accept the manuscript after the following minor revisions.

TITLE: Adequate.

ABSTRACT:  Adequate.

KEYWORDS: Redundant with the title (i.e. Aflatoxins can be changed with mycotoxins)

  1. INTRODUCTION: clear, well written. I would just suggest to use the same format for all units (e.g. always µg kg-1 or µg/kg) and to change Figure 2 with a better quality one. In line 144 you stated that Table 2 was proposed by Matabaro et al. (2017) but I cannot find the table or something similar in that paper. Was the reference wrong? Please explain.
  2. A. flavus AND AFLATOXIN PRODUCTION: adequate.
  3. FACTORS AFFECTING AFLATOXIN CONTAMINATION IN CROPS: this paragraph seems to be to synthetic with respect to the chapters 1 and 2. I would suggest to include some other information, i.e. relationship between climate change and aflatoxin occurrence in crops.
  4. AFLATOXINS MANAGEMENT IN CROPS: L356 please delete the sentence and start directly with 4.5.1. L424-428 please rephrase the sentence.

Author Response

Comments and Suggestions for Authors

GENERAL COMMENTS:

The paper deals with an interesting topic of clear interest for the readers of the Journal of Fungi. The paper presents a review of the biocontrol of Aflatoxins using non-aflatoxigenic A. flavus.  The review presents an adequate structure and an adequate literature study (recent and valid). The use of the English language is correct and the text is easy to read and understand.

Overall, I suggest accepting the manuscript after the following minor revisions.

TITLE: Adequate.

ABSTRACT:  Adequate.

KEYWORDS: Redundant with the title (i.e. aflatoxins has been changed to mycotoxins)

  1. INTRODUCTION: clear, well written. I would just suggest using the same format for all units (e.g. always µg kg-1 or µg/kg) and changing Figure 2 with a better quality one.

Ans: The format for all units has been standardized with µg/kg format and Figure 2 has been changed by the better quality one.

  1. In line 235, you stated that Table 2 was proposed by Matabaro et al. (2017) but I cannot find the table or something similar in that paper. Was the reference wrong? Please explain.

Ans: Sorry for the wrong reference. We have updated it with the following correct reference now.

Lizárraga-Paulín, E.G.; Moreno-Martínez, E.; Miranda-Castro, S.P. Aflatoxins and their impact on human and animal health: an emerging problem. Aflatoxins-Biochemistry and Molecular Biology2011, 13, 255-282.

  1. A. flavus AND AFLATOXIN PRODUCTION: adequate.
  2. FACTORS AFFECTING AFLATOXIN CONTAMINATION IN CROPS: this paragraph seems to be too synthetic concerning chapters 1 and 2. I would suggest including some other information, i.e. relationship between climate change and aflatoxin occurrence in crops.

Ans: A paragraph about climate change and aflatoxins occurrence has been added as suggested by the reviewer. 

  1. AFLATOXINS MANAGEMENT IN CROPS: L356 please delete the sentence and start directly with 4.5.1. L424-428 please rephrase the sentence.

Ans: L356 has been deleted.

         L424-428, the section related to CO2 impacts on efficacy of biocontrol agents (AF-) has been removed as per another reviewer comment.

Reviewer 3 Report

Food safety and plant protection for human health security is a major concern since many years; in particular, the containment of toxic metabolites such as mycotoxins is deserving the widest, coordinated effort from both the international Agencies and single Governments. Amongst the possible strategies, the prevention in field and during storage of the mycotoxigenic species infection is one of the most applicable (and applied), even if it is still mainly reached with the use of chemical pesticides (insecticides and fungicides). However the use of biocontrol agents, whose use increases every day, is considered the most desirable due to the reduction of chemical, toxic resudes in food and feed commodities. Hence, a review on the use of atoxigenic strains of A. flavus as intraspecific biocompetitors, effective in control the aflatoxin contamination and diffusion, is highly interesting and demanding.

Unfortunately, the present work does not satisfy this need... I found it badly balanced in terms of structure and organization: with regard to the title, that recalls a review focused on the biological control of aflatoxin contamination, an excessive space has been given to general informations about aflatoxin and A. flavus features; however, despite the huge amount of quality literature on this issues, the present is superficial and poorly informative, nontheless often imprecise and not updating at all. On the other hand, the part actually dedicated to the biocontrol is too scarce: many paragraphs are focused on topics related to biocontrol efficacy (such as the role of abiotic stresses) but without any real discussion about their affection on biocontrol strategies; additionally, paragraph 4.6.3 (Carbon Dioxide and Growth Non-aflatoxigenic Strains) has no sense at all....where is the link between the use of non-toxigenic strains used for aflatoxin biocontrol and Carbon dioxide? References ar not adequate and deepen enough. Actually, the paper appears as a mere collage of sparse pieces from the relevant literature, without any improvement of the state of the art nor any significant contribution to the field.

Author Response

Comments and Suggestions for Authors

Food safety and plant protection for human health security is a major concern for many years; in particular, the containment of toxic metabolites such as mycotoxins deserves the widest, coordinated effort from both the international Agencies and single Governments. Amongst the possible strategies, the prevention in the field and during storage of the mycotoxigenic species infection is one of the most applicable (and applied), even if it is still mainly reached with chemical pesticides (insecticides and fungicides). However, the use of biocontrol agents, whose use increases every day, is considered the most desirable due to reducing chemical, toxic residues in food and feed commodities. Hence, a review on the use of atoxigenic strains of A. flavus as intraspecific biocompetitors, effective in control aflatoxin contamination and diffusion, is highly interesting and demanding.

Unfortunately, the present work does not satisfy this need...

I found it badly balanced in terms of structure and organization: concerning the title, that recalls a review focused on the biological control of aflatoxin contamination, and excessive space has been given to general information about aflatoxin, and A. flavus features; however, despite the huge amount of quality literature on this issues, the present is superficial and poorly informative, nonetheless often imprecise and not updating at all. On the other hand, the part dedicated to the biocontrol is too scarce: many paragraphs are focused on topics related to biocontrol efficacy (such as the role of abiotic stresses) but without any real discussion about their affection on biocontrol strategies; additionally,

Ans: The manuscript structure and organization have been modified; the first section regarding aflatoxin has been rewritten with updated and good quality literature. The mycotoxins part, which was not closely related to the current work, has been removed. A real discussion has been added for each part, and hopefully, the current work will satisfy you.

Paragraph 4.6.3 (Carbon Dioxide and Growth Non-aflatoxigenic Strains) has no sense at all....where is the link between the use of non-toxigenic strains used for aflatoxin biocontrol and Carbon dioxide?

Ans: The paragraph related to Carbon Dioxide and Growth of Non-aflatoxigenic Strains has been removed as suggested by you.

References are not adequate and deepen enough. The paper appears as a mere collage of sparse pieces from the relevant literature, without any improvement of state of the art nor any significant contribution to the field.

Ans: Along with updated literature, the number of references has been increased from 114 to 164. All the changes we made in this manuscript are highlighted. We hope that our current effort will resolve your concern about manuscript quality and structure and satisfy you this time. Thank you so much for taking your precious time

Round 2

Reviewer 3 Report

When I reviewed the manuscript for the first time, I suggested to not publish it due of heavy concerns to the overall structure and organization of information. In they answers, Authors stated to have followed my comments, amending in a substantial way the main text. Unfortunately, the present version does not satisfy the basic requirements for a critical review on the use of non-toxigenic Aspergillus strains for the biocontrol of aflatoxin contamination in crops. This topic has been deepen in various and recent literature reviews, and, in order to provide the most updated and outstanding information to the audience of Journal of Fungi, a more accurate work should be done.

I found the manuscript still badly balanced in terms of structure and organization: despite my observation, an excessive space has been given AGAIN to general informations about aflatoxin and A. flavus features with regard to the title. Authors stated to have deleted the non relevant paragraphs, but this didn't happened. Chapter on Aflatoxin Effects on Human Health is still too lenghty, and must be shortened because the topic is limitedly relevant to the title (Biocontrol with atoxigenic strains).

The literature, commented as superficial, poorly informative and not updating at all, has only been augmented, but not improved.

Actually, the paper still appears as a mere collage of sparse pieces from the relevant literature, without any improvement of the state of the art nor any significant contribution to the field.

Line 33: "Aspergillus nomiae" must be changed in "Aspergillus nomius"

Line 38: "physiological procedures" it has no sense

Line 40: "sporadic incidence" is highly incorrect! Aflatoxins' incidence in food and feed commodities in a number of Countries is far from being sporadic

Figure 2 is useless.

Figure 3: if the scheme is taken from another paper, as the citation suggests, it should be stated in the caption.

Author Response

Review Report Form

Comments and Suggestions for Authors

When I reviewed the manuscript for the first time, I suggested not publish it due to heavy concerns to the overall structure and organization of information. In the answers, the Authors stated to have followed my comments, amending substantially the main text. Unfortunately, the present version does not satisfy the basic requirements for a critical review on the use of non-toxigenic Aspergillus strains for the biocontrol of aflatoxin contamination in crops. This topic has been deepened in various and recent literature reviews, and, to provide the most updated and outstanding information to the audience of the Journal of Fungi, more accurate work should be done.

I found the manuscript still badly balanced in terms of structure and organization: despite my observation, an excessive space has been given AGAIN to general information about aflatoxin and A. flavus features concerning the title. The authors stated to have deleted the non-relevant paragraphs, but this did not happen. The chapter on Aflatoxin Effects on Human Health is still too long and must be shortened because the topic is limitedly relevant to the title (Biocontrol with atoxigenic strains). The literature, commented as superficial, poorly informative, and not updating at all, has only been augmented, but not improved. The paper still appears as a mere collage of sparse pieces from the relevant literature, without any improvement of the state of the art nor any significant contribution to the field.

Part 1:

  • The introduction has been shortened from 8 pages to only 1 and a half pages, removing all irrelevant heading, including Chemical Structure of Aflatoxins, Toxicity of Aflatoxins, Aflatoxin Effects on Human Health, Acute Toxicity, Chronic Toxicity, Impacts on Animals Health, and Aflatoxins Exposure.
  • Tables and Figures related to the parts have also been removed.
  • A new introduction has been written which is more relevant to the Title, giving a background to aflatoxins.
  • The introduction part contains updated literature, informative and concise to the title
  • The references in this part have been updated too.

Part 2:

  • The literature Aspergillus flavus and aflatoxin portion have been shortened into only one paragraph and the heading has been modified into Global Distribution of Aspergillus flavus and aflatoxins.
  • The subheading, like variability in Aspergillus flavus, Morphology, Genetic Variability, Molecular Differences, Virulence have been removed.
  • Figure 4. Schematic presentation of A. flavus life cycle on corn has been removed too.

Line 33: "Aspergillus nomiae" must be changed in "Aspergillus nomius"

  • Line 34, and 77: The word Aspergillus nominae has been changed into Aspergillus nomius.

Line 38: "Physiological procedures" has no sense

  • Line 40: The sentence has been revised and rewritten.

Line 40: "sporadic incidence" is highly incorrect! Aflatoxins' incidence in food and feed commodities in several Countries is far from being sporadic.

  • Line 40: The sentence is deleted as the introduction part has been updated.

Figure 2 is useless.

  • Figure 2: As we updated and shortened the introduction part, all the figures have been deleted.

Figure 3: if the scheme is taken from another paper, as the citation suggests, it should be stated in the caption.

  • Figure 3: has been deleted too.

Overall, we have reduced the manuscript from 20 pages to only 13 pages, removing all the irrelevant headings, tables, and figures. The literature has been updated will deep and updated references. The structure of the manuscript has been restructured and well organized.

Thanks.
